# HeLoM: Progressive Disease Detection with Heterogeneous and Longitudinal EHRs via Memory-Augmented LLMs

## Abstract

Recent developments in large language models (LLMs), have significantly advanced healthcare applications, especially the electronic health record (EHR) processing, and demonstrated great potential in disease prediction. EHR are digital records of patients' medical data, including historical visits, diagnoses, lab tests, and treatments, organized across hospital visits for clinical and research use. Despite LLMs' great potentials, previous methods to predict disease with EHRs based on LLMs face several persistent challenges: (1) they often concatenate short and fixed number of EHR visits (e.g., the latest five) from individual patients and then feed it to LLMs due to either limited input context length or LLMs' capabilities to understand long context, which limits the disease prediction with longitudinal EHR; (2) most prior work focuses on clinical note and overlook EHR's inherent nature like heterogeneity; and (3) EHR are characterized by heterogeneous patterns of missingness (e.g., the missingness of various vital signs). To tackle these problems, we propose a novel progressive memory-augmented framework HeLoM that consists of three key steps: For the first challenge, in a current EHR visit, HeLoM first adaptively fetches previously refined memory (i.e., the patient's previous visits) most relevant to the current disease prediction and then refine this visit to update its memory bank. For the second challenge, we incorporate the heterogeneous data, vital signs, from EHR to enhance the prediction performance. For the third challenge, we introduce two imputation strategies to handle missing data: one leverages LLMs to generate plausible values, and the other applies linear interpolation algorithms to estimate the missing value. By collecting a real-world longitudinal EHR data on Type-2 diabetes from the hospital of our institution, we show the superior performance of HeLoM in disease prediction in terms of both prediction accuracy and early detection. Comprehensive ablation studies underscore the importance of generating missing values from heterogeneous sources, and provide insights into building reliable systems for real-world EHRs.

## 1 Introduction

The integration of artificial intelligence (AI) into healthcare, particularly through large language models (LLMs) such as GPT-4 (Achiam et al., 2023) and DeepSeek-R1 (Guo et al., 2025), has profoundly transformed modern health management. Emerging research highlights the potential of LLMs to enhance clinical decision-making through the use of electronic health records (EHRs) (Li et al., 2024). EHRs are comprehensive, longitudinal, and digital repositories of patients' health information created and maintained in clinical settings. EHRs are characterized by their richness and scale, but also by challenges such as incompleteness, heterogeneity, irregular sampling across visits, and variability in clinical documentation. These features make EHRs both a powerful resource for predictive modeling, and a complex domain requiring careful data preprocessing and methodological adaptation (Yang et al., 2022; Theodorou et al., 2023). EHRs often span decades of patient history, which makes the data not only large in volume but also longitudinal in nature, containing sequential observations across irregular time points. Therefore, this rich, heterogeneous, and longitudinal data offers both immense potential and significant challenges for disease modeling and prediction (Bush et al., 2017; Li et al., 2020).

To fully harness the potential of LLMs for disease prediction using EHRs, several key challenges must be addressed. **First**, EHR data is inherently longitudinal and often contains lengthy clinical notes. Existing methods typically limit input to a fixed number of visits (e.g., the most recent five) due to resource constraints, the context length limitations of LLMs, or the models' ability to reason over long sequences. Clinical notes from these visits are concatenated and fed into the model, which restricts adaptability. For example, LLMs may require more (or fewer for early detection) patients' visits to capture sufficient diagnostic context and preserve continuity across time (Hager et al., 2024; Zhu et al., 2024). **Second**, EHRs encompass diverse heterogeneous data, including structured measurements such as vital signs (e.g., temperature, weight, body mass index) recorded at each visit, alongside unstructured clinical notes. This heterogeneity and irregularity complicate integration and necessitate robust strategies to extract clinically meaningful information. **Third**, EHRs are characterized by heterogeneous patterns of missingness. Prior work has highlighted the importance of missing data in healthcare and emphasized the need for principled methods to address it (Zhou et al., 2025).

To jointly address these challenges, we first collect heterogeneous and longitudinal EHR data from Type-2 diabetes (T2D) patients and the control group (non-T2D) spanning 10 years in EPIC systems from a hospital. Motivated by prior evidence that memory mechanisms can help LLMs store and retrieve information efficiently (Liang et al., 2024; Zhang et al., 2023), we design an adaptive memory-augmented framework that addresses the unique challenges of heterogeneous and longitudinal EHR data. Specifically, each patient visit is iteratively fed into the LLM. For the current visit, LLMs will utilize the past visits from the "memory bank" that are important for the current prediction as the context. This combined input (past memory and the current patient visit) is then fed into the LLM to predict disease. Unlike previous methods, these settings enable LLMs to adapt to different patients, where different numbers of visits are dynamically used to ensure better disease prediction. It also helps effectively expand the usable context length for the EHRs, since the input will only include the current visit and a refined important prior visits. In addition, to deal with the heterogeneous data, we incorporate the patient's vital signs from each visit as supplementary information. However, these vital signs are often incomplete, resulting in missing or irregular values that complicate reliable modeling. To handle the incomplete vital signs, we explore two complementary imputation strategies: (1) leveraging the reasoning capabilities of the LLM to infer and complete the missing values. (2) applying an linear interpolated imputation workflow. Taken together, we contribute to the following directions:

- We propose a novel memory-augmented inference-based framework, referred to as *HeLoM*, that enables LLMs to perform iterative disease prediction without any fine-tuning or gradient updates. Our approach dynamically refines each patient visit note one at a time and accumulates the previously refined notes in a memory bank through a step-by-step pipeline, supporting efficient inference across varying visit lengths.

- We introduce a heterogeneity-aware prompting mechanism that incorporates incomplete structured data (e.g., vital signs) with clinical notes. Then we employ two complementary imputation strategies: leveraging the LLM reasoning ability and classic interpolated imputation methods to generate the missing values. These approaches improve prediction robustness, particularly in handling heterogeneous and missing data.

- Due to the limited number of complete longitudinal EHRs available to the public, we collect a real-world EHR dataset spanning 10 years (2011-2021) from a hospital for T2D. Empirical analyses show that our methods achieve superior performance in disease prediction with earlier diagnosis visits, particularly under data scarcity, imbalanced visit frequency, and heterogeneous EHR settings.

## 2 RELATED WORKS

Recent advances in LLMs has opened up new possibilities for healthcare applications. Notably, GPT-4 has demonstrated exceptional performance in answering medical questions, showcasing strong zero-shot and few-shot reasoning capabilities on expert-level content (Nori et al., 2023a). Building on this, Medprompt (Nori et al., 2023b) introduces specialized prompting strategies that enhance the effectiveness of general foundation models like GPT-4. For medical LLMs, Yang et al. (2022) introduced GatorTron, the first clinical LLM trained from multi-billion parameter scale,

achieving substantial improvements over previous general methods. Google's Med-PaLM was the first LLM to reach the passing threshold on the US Medical Licensing Examination (USMLE) (Singhal et al., 2022). Its successor, Med-PaLM 2, combined a more powerful base model (PaLM 2), medical domain-specific fine-tuning, and tailored prompting strategies. It achieved expert-level performance, received higher ratings from physicians for quality in answering patient questions (Singhal et al., 2025). In addition, while LLMs demonstrate strong potential for handling heterogeneous EHR data, only a limited number of methods have been proposed to effectively integrate these diverse sources. For example, Zhang et al. (2024) proposed a heterogeneous mixture of LoRA expert modules that aggregate architecturally diverse models. Similarly, Liu et al. (2025) extended LoRA by incorporating rank heterogeneity to enhance both communication and computational efficiency. Another line of research focuses on managing heterogeneity of data. Wang et al. (2025a) introduced semantic operators to enable heterogeneous data analytics in data lakes, while Zhang et al. (2025) proposed two novel disaggregation techniques to jointly address model and data heterogeneity. Efforts have also been made to leverage knowledge graphs and graph neural networks (GNNs): Ko et al. (2024) exploited knowledge graphs to fill missing values, and Gao et al. (2025b) employed GNNs to encode complex relationships within heterogeneous data. In addition, Gao et al. (2025a) combined LLM-based summarization and classification with GNN-based representation to handle diverse data. Several works studied domain-specific scenarios where heterogeneity is salient. For instance, Wang et al. (2025b) introduced a temporal batching prediction framework to improve efficiency in medical Q&A. More broadly, Tang et al. (2024b) developed a large graph model with a heterogeneous graph instruction tuning paradigm to capture complex relational heterogeneity. While these approaches demonstrate promising directions, they often require complex architectures, typically substantial labeled data, and computational resources. Furthermore, it has yet to show robustness in heterogeneous and relatively large real-world EHR datasets.

Besides these, longitudinal EHRs present unique challenges due to their temporal complexity. In handling longitudinal clinical data, LLMs have shown promising capabilities in modeling temporal sequences, and interpreting medical texts, highlighting their applicability in health care Loni et al. (2025). For example, Thompson et al. (2023) utilized Retrieval-Augmented Generation (RAG) (Lewis et al., 2020) to retrieve the related clinical records. The text is then used to infer the hospital stays and ICU episodes tasks. Pellegrini et al. (2025) proposed feature-wise and visit-wise modeling approaches to capture the temporal structure of longitudinal EHRs. However, they only use regular expressions to map the related disease and then aggregate the results. Tang et al. (2024a) assigned specialized LLM agents to extract signs and symptoms relevant to AD and conducted domain-specific evaluations. Cui et al. (2025) leveraged a multi-agent framework, where a predictor and a critic agent are used to enhance reasoning and prediction performance. These methods have shown improvements over previous methods, however, such a setup incurs substantial computational and latency costs, which may limit its practicality in real clinical settings. Recent work has also leveraged longitudinal EHRs with LLMs to support the detection of pancreatic cancer (Park et al., 2025). However, the authors note that features within the 0-3 month diagnostic window carry disproportionately strong signals, raising concerns of optimistic performance estimates. Furthermore, these methods often come with a high cost of computational resources and careful design of a module to model the features. We therefore introduce HeLoM, a memory-augmented framework that enables LLMs to iteratively refine patient visits without fine-tuning. It also combines heterogeneity-aware prompting with two different imputation strategies to enhance performance.

## 3 METHODS

HeLoM aims to address: (1) difficulty in adaptively capturing long and irregular temporal sequences across heterogeneous modalities, (2) limited scalability due to the context length restrictions of LLMs, and (3) insufficient strategies for handling pervasive missing data in clinical records.

### 3.1 FRAMEWORK OVERVIEW

HeLoM (**Figure** 1) builds on the idea of adaptive memory–augmented inference for longitudinal EHR. Unlike conventional approaches that either truncate patient history to a few visits or naively concatenate all encounters into a single context, both of which risk losing critical temporal dependencies or overwhelming the model's input window, our method relies on a lightweight and evolving

external memory. This memory is iteratively refined during test time, ensuring that information most relevant for disease prediction is retained while redundant or outdated details are pruned. The framework operates as a cyclic pipeline composed of three tightly coupled modules:

- **Generation of missing structured data**: incomplete vital signs are completed either by LLM-based synthesis, which generates plausible values in context, or by linear interpolation for continuous signals. The structured data are then formatted through text templates and appended after clinical notes. The integration is used for disease prediction and memory update.

- **Disease prediction with "memory bank"**: each new patient visit is processed jointly with curated past memory, current visit and the imputed vital signs, enabling the model to leverage longitudinal continuity while adapting to variable visit lengths.

- **Memory refinement**: after each prediction, the model synthesizes the current encounter and existing memory into a compact representation that will guide subsequent inferences.

This iterative process allows the framework to co-evolve its predictions and memory representations under a single LLM over time, effectively bridging the gap between zero-shot inference and the cumulative reasoning needed in longitudinal EHR analysis.

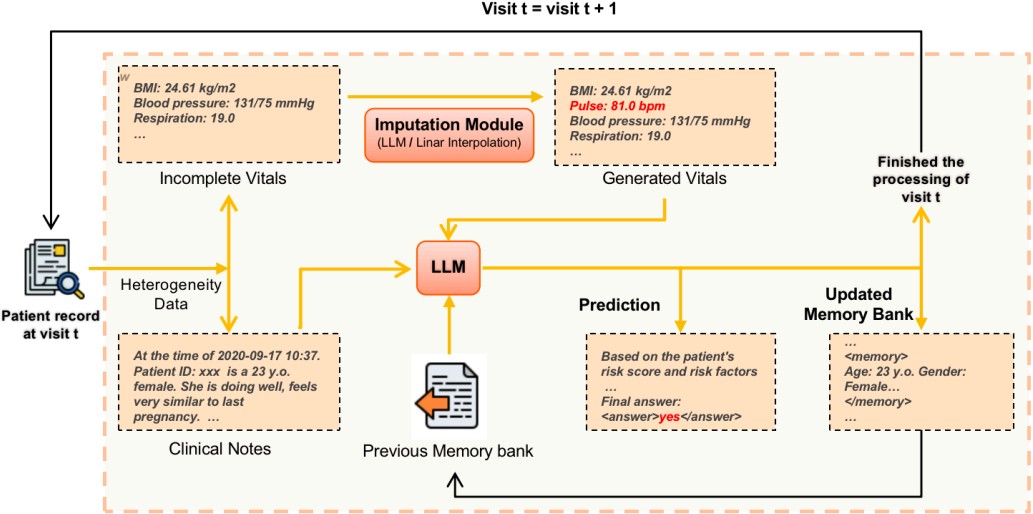

Figure 1: Pipeline of the Proposed Framework HeLoM.

## 3.2 GENERATION OF MISSING STRUCTURED DATA

Clinical records such as vital signs are inherently structured and usually incomplete, with values missing due to irregular documentation practices or measurement errors. In this study, we focus on a standard set of commonly recorded vital signs for T2D in EHRs and hospital settings, including *Temperature, Weight, FiO2, Pain, BMI, SpO2, and others*. Different from prior work that primarily relies on clinical notes (Hager et al., 2024; Zhu et al., 2024), HeLoM explicitly leverages these structured data by imputing missing values through context-aware strategies and integrating them into prompts for downstream prediction and memory update. The detailed prompt design for incorporating heterogeneous data is provided in Appendix C. For data missingness, our framework examines two imputation mechanisms to ensure reliability. Formally, let the vital sign vector $\mathbf{v}_t$ at visit $t$ be:

$$\mathbf{v}_t = \left[ v_t^{(\text{BMI})}, v_t^{(\text{BP})}, v_t^{(\text{SpO}_2)}, \dots \right], \tag{1}$$

where missing entries occur irregularly across visits. We consider two strategies for imputing the missing values: (1) **LLM-based imputation**. Leveraging the reasoning capacity of the LLM, missing values of the vital signs are generated by conditioning on the current incomplete values, clinical note, and the LLM's memory bank (Sec. 3.4). Based on that, we can finalize the formula:

$\tilde{v}_t = \text{GD}(v_t, x_t, \tilde{m}_{t-1})$, where GD, $x_t$, and $\tilde{m}_{t-1}$ stand for "Generate Data", current visit note, and previous memory, respectively. The imputed values are appended to the encounter note in a standardized format, ensuring that structured and unstructured modalities are jointly available for downstream prediction. This approach captures temporal continuity and patient-specific patterns, yielding values that are both clinically plausible and contextually consistent. The prompt can be accessed in Appendix C (2) **Interpolation-based imputation**. Each patient's data is conceptualized as a matrix $V$, where an element $v_{t,j}$ corresponds to the $j$-th vital sign measured during the $t$-th visit. When a value $v_{t,j}$ was missing, it was imputed based on the temporally adjacent non-missing values for that same vital sign. The estimated value, denoted as $\hat{v}_{t,j}$, was calculated using the following formula:

$$\hat{v}_{t,j} = v_{t_p,j} + \frac{t - t_p}{t_f - t_p} \cdot (v_{t_f,j} - v_{t_p,j}) \tag{2}$$

In this equation, $v_{t_p,j}$ represents the last valid observation of the vital sign prior to the missing entry (at visit $t_p$), and $v_{t_f,j}$ is the next subsequent valid observation (at visit $t_f$). The term $\frac{t - t_p}{t_f - t_p}$ serves as a temporal weight, ensuring that the imputed value is proportionally influenced by its proximity to the two neighboring data points. This method preserves the underlying temporal trend within the series for each vital sign.

### 3.3 Progressive Disease Prediction with Memory Bank

Given an $t$-th EHR visit $x_t = (x_{t,1}, x_{t,2}, \dots, x_{t,n})$ from a patient's EHRs $\mathcal{D}_{\text{patient}}$, after imputing missing vital signs with information from both the current encounter and previously refined memory, our framework integrates three complementary sources for early disease prediction: (1) the imputed set of vital signs $\tilde{v}_t$ from the current visit, (2) raw clinical notes and structured features from the current encounter (e.g., lab values, medications), and (3) the memory distilled from prior visits in the LLM's "memory bank", where each refined note is sequentially stored and cumulatively retrieved to inform subsequent predictions. The prediction of $t$-th visit $pred_t$ can be formulated as:

$$\tilde{pred}_t = LLM(x_t, \tilde{v}_t, \tilde{m}_{t-1}), \tag{3}$$

where $\tilde{m}_{t-1}$ is the previous memory. These components are seamlessly combined through a pre-defined template (See Appendix C) and fed into the LLM to determine whether the patient has the target disease. This design ensures adaptability to both cold-start patients (with no prior memory, $\tilde{m}_0 = \varnothing$) and longitudinal scenarios, where multiple visits are available, predictions benefit from the accumulated context encoded in $\tilde{m}_{t-1}$. These prompts can be seen in Appendix C. Unlike approaches that rely on continuous retraining, our method performs test-time adaptation purely through contextual enrichment. This strategy avoids expensive gradient updates, while still enabling the system to adjust its reasoning across sequential inputs. By reusing distilled longitudinal context, the framework reduces repetitive errors and provides more consistent predictions across time. Further, the structured design allows seamless incorporation into clinical pipelines. Since predictions are generated without retraining, they can be continuously updated as new visits occur, supporting real-time monitoring in EHR systems.

### 3.4 Progressive Update of Memory

One key step is to update memory dynamically after each visit. Rather than simply appending all past notes, which quickly leads to input overload, HeLoM synthesizes a concise, evolving representation that captures salient information for future predictions. At each visit $t$, the new memory state is generated as using the same LLM from Secs. 3.2 and 3.3:

$$\tilde{m}_t = ME(x_t, \tilde{v}_t, \tilde{m}_{t-1}), \tag{4}$$

where $ME(\cdot)$ denotes the process of Memory Evolution with structured vitals, unstructured notes, and the prior memory. Inspired by Dynamic Cheatsheet (Suzgun et al., 2025), this refinement is guided by the format prompt and the following three principles: (1) *Contextual relevance*: prioritizing information that is essential for longitudinal reasoning, such as abnormal trajectories or recurring symptoms, while discarding redundant or short-lived fluctuations. (2) *Compactness*: compressing visit-level evidence into structured descriptors or concise textual snippets, ensuring the evolving memory remains manageable and retrievable. (3) *Adaptivity*: incorporate predictive cues identified

at inference (e.g., whether a certain lab combination strongly influenced the model's decision) so that future predictions can reuse these insights. In practice, the refinement step operates as a two-way filter: it selects useful information from the current visit and simultaneously re-evaluates older memory content. Outdated or less relevant patterns can be down-weighted or pruned, preventing the accumulation of irrelevant noise. This ensures that the memory reflects a balanced, high-yield summary rather than a chronological archive. The prompt can be seen in Appendix C.

The iterative refinement mechanism is especially advantageous in EHR data, where heterogeneity and redundancy are pervasive. First, repeated lab tests across visits may produce overlapping information, while narrative notes often contain boilerplate text. By distilling these inputs into concise, context-aware memory states, the model avoids being overwhelmed by irrelevant repetition. Another benefit of this design is that it creates a feedback loop between prediction and memory update. Each prediction not only generates an outcome $\tilde{pred}_i$ but also leaves a trace in the evolving summary. Over time, this co-adaptation ensures that the system becomes more aligned with both patient-specific patterns and broader population-level regularities, showing potential for *early* disease detection. Finally, from a systems perspective, this refinement allows the framework to scale efficiently. Since the evolving memory remains bounded in size and updated iteratively, the model can be deployed in real-world clinical settings without exceeding computational budgets or context-window limitations. This provides a practical pathway for continuous monitoring of chronic conditions such as diabetes.

## 4 EXPERIMENTS

### 4.1 SETUP

**Dataset.** Due to the limited full metadata for EHRs in the public, we curated a longitudinal EHR dataset in collaboration with the hospital of our institution to support T2D research. The cohort comprises 354 adult patients (over 18 years old) who received care between 2010 and 2021, each with at least two outpatient visits within 24 months. The dataset has roughly half patients diagnosed with T2D and the other half serving as the control group. The Avg Disease Onset is the average visit index when an adult is detected as T2D. The dataset

Table 1: Patient Visits and Disease Onset for T2D Dataset.

| Statistic | Value |
|---|---|
| Avg #Visit | 6.15 |
| Max #Visit | 44 |
| Min #Visit | 3 |
| Median #Visits | 4 |
| Avg Disease Onset | 5.33 |

consists of vital signs and free-text clinical notes, and visit dates. Notes that are shorter than 150 words or those explicitly containing T2D diagnoses were excluded. Laboratory measures in the notes include glucose, HbA1c, lipid profiles, and complete blood counts, while vital signs cover BMI, blood pressure, pulse, respiration, and others. This heterogeneous dataset provides a foundation for developing predictive models of T2D from longitudinal EHRs. Some basic statistics of the T2D dataset are shown in Table 1, and more details of the dataset can be seen in Appendix B.

**Baselines and Backbone LLMs.** We compare our approach with (1) the common paradigm that concatenates all available visit records of a patient as much as the model's context length allows in chronological order into a single textual input, preserving the temporal sequence of clinical information. The resulting text is then fed into LLMs with a context window of either 16k or 24k tokens. The models are prompted to analyze the patient's clinical history and directly generate a diagnosis-related response (e.g., whether the patient has T2D); and (2) PromptEHR (Zhu et al., 2024) which designs a new prompting-based approach for structured longitudinal EHR data, flexibly modeling variable numbers of visits. The input also includes both structured features and free-text.

We utilized a diverse range of mainstream LLMs as the backbone prediction models, including DeepSeek-R1-0528-Qwen3-8B (Guo et al., 2025), Mistral-7B-Instruct-v0.2 (Jiang et al., 2023), Llama3.1-8B-Instruct (Grattafiori et al., 2024) and one medical LLM MediPhi-Instruct (Corbeil et al., 2025). MediPhi from Microsoft is fine-tuned on various medical corpora and specializes in the medical and clinical domains. They are noted as DeepSeek, Mistral, Llama3.1, and MediPhi for short, respectively. The decoding parameters were set as follows: temperature = 0, and top-$p$ = 1.0. All other parameters, including top-$k$ and repetition penalty, were kept at their default value. Inference processing was under the vLLM library[1]. All the prompts used can be found in Appendix C.

---

[1]https://github.com/vllm-project/vllm

**Metrics.** To evaluate HeLoM, we use the following metrics: (1) classification metrics for disease prediction, including Accuracy, Precision, Recall, and F1-score; (2) Avg PV: the average visit index when the model predict the disease. This is to examine HeLoM's potential in early disease prediction.

## 4.2 MAIN RESULTS

We evaluate five prompting paradigms: Baseline-16K, Baseline-24K, PromptEHR, and HeLoM with two data imputation methods for generating the missing values, denoted as HeLoM w. LLM's V for using LLM and HeLoM w. Interp's V for using linear interpolation. The best and second-best results are in bold and underlined font in Table 2, respectively.

Table 2: Performance of models in different prompting paradigms. BL stands for the baseline.

| Model | Variant | Accuracy | Precision | Recall | F1-score |
|---|---|---|---|---|---|
| DeepSeek | BL-16K | 0.579 | 0.679 | 0.299 | 0.416 |
| | BL-24K | 0.590 | 0.705 | 0.311 | 0.431 |
| | PromptEHR | **0.672** | 0.665 | 0.695 | 0.680 |
| | HeLoM w. LLM's V | 0.638 | 0.592 | 0.893 | **0.712** |
| | HeLoM w. Interp's V | 0.588 | 0.551 | 0.949 | 0.697 |
| | PromptEHR w/o V | 0.623 | **0.770** | 0.337 | 0.469 |
| | HeLoM w/o V | 0.554 | 0.530 | **0.955** | 0.681 |
| Mistral | BL-16K | 0.672 | 0.814 | 0.446 | 0.577 |
| | BL-24K | 0.658 | **0.841** | 0.390 | 0.533 |
| | PromptEHR | 0.623 | 0.770 | 0.337 | 0.469 |
| | HeLoM w. LLM's V | **0.701** | 0.696 | **0.712** | **0.704** |
| | HeLoM w. Interp's V | 0.692 | 0.705 | 0.661 | 0.682 |
| | PromptEHR w/o V | 0.625 | 0.770 | 0.339 | 0.471 |
| | HeLoM w/o V | 0.667 | 0.695 | 0.593 | 0.640 |
| Llama3.1 | BL-16K | 0.590 | 0.670 | 0.356 | 0.465 |
| | BL-24K | 0.607 | **0.732** | 0.339 | 0.463 |
| | PromptEHR | 0.497 | 0.476 | 0.057 | 0.101 |
| | HeLoM w. LLM's V | **0.669** | 0.622 | 0.864 | 0.623 |
| | HeLoM w. Interp's V | 0.613 | 0.569 | **0.927** | 0.705 |
| | PromptEHR w/o V | 0.497 | 0.476 | 0.057 | 0.101 |
| | HeLoM w/o V | 0.664 | 0.614 | 0.881 | **0.724** |
| MediPhi | BL-16K | 0.610 | 0.617 | 0.582 | 0.599 |
| | BL-24K | 0.590 | 0.595 | 0.565 | 0.580 |
| | PromptEHR | 0.521 | 0.509 | **0.846** | 0.636 |
| | HeLoM w. LLM's V | 0.621 | 0.606 | 0.695 | **0.647** |
| | HeLoM w. Interp's V | 0.596 | 0.583 | 0.672 | 0.625 |
| | PromptEHR w/o V | **0.623** | **0.770** | 0.337 | 0.469 |
| | HeLoM w/o V | 0.569 | 0.559 | 0.648 | 0.600 |

We can observe that HeLoM markedly improve Recall and F1-score, while maintaining competitive levels of Accuracy and Precision. For instance, with DeepSeek as the backbone LLM, HeLoM achieves the highest F1-score of 0.712, which substantially surpass other methods. This highlights that, while baselines (including PromptEHR) often achieve relatively high Precision, they suffer from lower Recall. By contrast, our methods strike a better balance between Precision and Recall, yielding stronger F1-scores. In disease prediction, improvements in Recall are particularly meaningful: failing to detect high-risk patients (false negatives) is far more detrimental than producing additional false positives. An additional insight is that the LLM-based imputation method often achieves the better Recall and F1-score across models, while still maintaining competitive Precision. In Llama3.1 with PromptEHR, we observe a huge drop in Recall and F1-score, because 94% of its prediction is non-T2D. For MediPhi, although PromptEHR attains the highest Recall (0.846), it does so at the cost of markedly reduced Precision and overall accuracy, same for DeepSeek. In contrast, HeLoM provides a more balanced performance profile across different backbone LLMs, as evidenced by consistently higher F1-scores (e.g., 0.647 in HeLoM with LLM's V vs. 0.636 in PromptEHR) in MediPhi. These findings suggest that the principled design of HeLoM —memory-augmented, heterogeneity, and missingness handling – enables it to achieve a more clinically desirable balance between sensitivity and reliability, offering more consistent identification of high-risk patients compared to baseline methods.

## 4.3 AVERAGE VISIT INDEX FOR DISEASE DETECTION

This experiment evaluates HeLoM's ability for early disease detection across longitudinal visits. At visit $t$, HeLoM has access to patient's records up to $t$ and then outputs a disease prediction; once the prediction is positive, the iteration stops and the corresponding visit index is recorded as the disease detection point. This setting mirrors realistic clinical monitoring, where patients are sequentially assessed until diagnosis becomes evident. Here, we do not compare with the baselines as the number of visits used for their disease prediction depends on the LLM's context length. Rather, we compare HeLoM with its variants–excluding vital signs and using original incomplete vitals (Ori V).

Table 3 shows the average visit index (Avg PV) for disease prediction, where a lower value indicates an earlier detection. HeLoM consistently predicts T2D at between the first and second visit on average; for instance, DeepSeek-HeLoM with interpolated vitals detects onset at 1.43 visits on average compared

Table 3: Average visit index for T2D detection of various models.

| Model | Methods for Vitals | ↓ Avg PV |
|-------|-------------------|----------|
| DeepSeek | Without V | 1.61 |
| | Original V | 1.86 |
| | LLM's V | 1.70 |
| | Interp's V | **1.43** |
| Mistral | Without V | 2.38 |
| | Original V | 2.06 |
| | LLM's V | 2.18 |
| | Interp's V | **1.99** |
| Llama3.1 | Without V | 1.55 |
| | Original V | **1.30** |
| | LLM's V | 1.50 |
| | Interp's V | 1.40 |
| MediPhi | Without V | 1.98 |
| | Original V | 1.99 |
| | LLM's V | **1.74** |
| | Interp's V | 1.76 |

to 1.86 with Ori V, while MediPhi-HeLoM achieves 1.74 visits using LLM-generated vitals. For Llama3.1, using the original incomplete vital signs yields earlier prediction results than HeLoM. This might be attributed to Llama3.1's internal capabilities of inferring the missing values even without data imputation. Together with results in Sec. 4.2, we show that HeLoM not only presents better prediction performance but also earlier prediction results.

## 4.4 ABLATION STUDY

**Impact of Data Imputation.** To study the impact of HeLoM's imputation methods, we further conduct a series of ablation experiments. We examined the following additional settings: (1) baselines (BL-24K and PromptEHR) with LLM-generated vital signs. We choose LLM-based over interpolation-based data imputation as the former generally achieves better performance. (2) HeLoM with original incomplete vital signs (HeLoM w. Ori V). By comparing performance across these various settings, we can assess whether imputation of vital signs can improve models' performance.

Table 4: Ablation Experiments For Imputation of Vital Signs.

| Model | Methods | Accuracy | Precision | Recall | F1-score |
|-------|---------|----------|-----------|--------|----------|
| DeepSeek | BL-24K + LLM's V | 0.588 | 0.559 | 0.831 | 0.668 |
| | PromptEHR + LLM's V | **0.658** | **0.639** | 0.729 | 0.680 |
| | HeLoM w. Ori V | 0.646 | 0.603 | **0.859** | **0.709** |
| Mistral | BL-24K + LLM's V | **0.689** | **0.825** | 0.480 | 0.607 |
| | PromptEHR + LLM's V | 0.658 | 0.630 | **0.746** | **0.683** |
| | HeLoM w. Ori V | 0.686 | 0.723 | 0.605 | 0.658 |
| Llama3.1 | BL-24K + LLM's V | 0.585 | 0.561 | 0.780 | 0.652 |
| | PromptEHR + LLM's V | 0.557 | **0.672** | 0.220 | 0.332 |
| | HeLoM w. Ori V | **0.632** | 0.618 | **0.844** | **0.713** |
| MediPhi | BL-24K + LLM's V | 0.559 | 0.565 | 0.514 | 0.538 |
| | PromptEHR + LLM's V | 0.506 | 0.500 | **0.876** | **0.637** |
| | HeLoM w. Ori V | **0.616** | **0.607** | 0.655 | 0.630 |

Combining results in Table 2 and Table 4, we can observe that with LLM-based data imputation, performance of all models including baselines is significantly improved. For Mistral, the gain of performance is more pronounced: For instance, PromptEHR with imputation enhances the Recall (0.746) and F1-score (0.683) over PromptEHR without imputation (Recall: 0.337, F1-Score: 0.469). These results suggest that data imputation plays a significant role in disease prediction with EHRs. Additionally, HeLoM still outperforms baselines with data imputation, suggesting the importance of "memory bank" for disease prediction with longitudinal EHRs.

**Impact of Incorporating Vital Signs.** To study the impact of incorporating vital signs, we perform additional experiments on HeLoM and PromptEHR where vital signs are removed from the input. As shown in Table 2, incorporating vital signs mostly improves model performance across prompting paradigms. For HeLoM, it achieves higher Recall or F1-scores than its variant that does not consider vital signs, particularly for DeepSeek, MediPhi and Mistral, where F1-score gains are substantial. The exception is for Llama3.1, which is partly due to its limited capabilities in handling heterogeneous data. Together, these findings un-

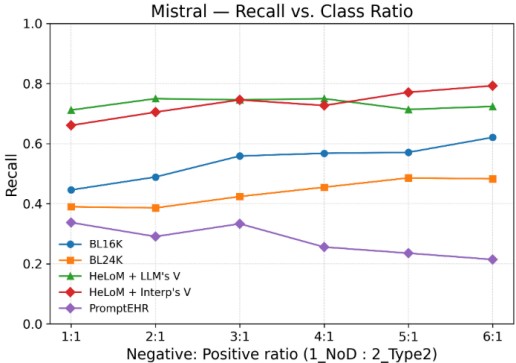

Figure 2: Mistral — Recall under different imbalance ratios.

derscore the importance of incorporating vital signs for reliable disease detection. Similar to PromptEHR, its performance, especially regarding Recall and F1 score, greatly degrades when vital signs are removed when DeepSeek and MediPhi are the backbone models. **Impact of Data Imbalance Ratio.** In practice, T2D prevalence is low; a system useful in clinical settings should maintain performance as positive samples become rare. Take Mistral as an example, by fixing the number of negative samples and randomly reducing positive samples, we test HeLoM's performance when varying the T2D:NonT2D ratio among {1:1, 1:2, ..., 1:6}. As shown in Figure 2, our methods remain high recall across all data ratios, clearly outperforming baselines. The proposed HeLoM with different data imputation methods yields consistently higher sensitivity to T2D samples, indicating better robustness to class imbalance.

## 4.5 ERROR CASE ANALYSES

To gain a deeper understanding of HeLoM's performance, we conducted an error analysis by randomly sampling 100 mis-classified examples from HeLoM w. LLM's V, using Llama 3.1 as an illustrative example. We chose Llama 3.1 because its results exhibited distinct patterns of behavior compared to the other backbone models. Specifically, we find two types of errors common for the Llama3.1 model: (1) failure to synthesize dynamically meaningful variations in the generated vital signs. This error type occupies 8% of the incorrect predictions. (2) unclear or partial synthesized data. Such an error occupies 14% of the incorrect predictions. This indicates that more advanced data synthesis methods are warranted for longitudinal and heterogeneous EHRs. See Appendix D for an example of each error type.

## 5 CONCLUSION

This work presents HeLoM, a progressive memory-augmented framework for disease prediction with longitudinal and heterogeneous EHRs collected from a real-world hospital. HeLoM addresses three central challenges: limited and non-adaptive context, heterogeneous data integration, and missingness handling. By iteratively updating memory bank, combining vitals with clinical notes, and applying both LLM-based and linear interpolation-based imputations, HeLoM consistently improves Recall and F1-scores across diverse backbone models–an especially valuable property for minimizing missed diagnoses in clinical settings. Our analyses further highlight the critical role of vital signs besides clinical notes and the benefits of generating them through imputation. Beyond these, HeLoM demonstrates robustness under class imbalance and achieves earlier disease detection than existing paradigms, underscoring its practical potential for real-world patient monitoring.

This study also has limitations that suggest promising future work. First, patient data and compute servers were kept offline for privacy reasons, preventing access to other cloud-based LLMs. Second, all experiments were conducted on a private, single-institution EHR dataset, though the use of EHRs is from real-world settings. Third, our study is focused on text and tabular data other than images. Future research should incorporate open datasets and develop multimodal foundation models to enhance capabilities of HeLoM.

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

## A    APPENDIX

This appendix provides supplementary materials that support the methodological framework and predictive analyses described in the main text. In particular, it includes detailed prompts, evolving reference structures, and illustrative examples that were used to guide diabetes risk prediction from EHRs. These materials are intended to demonstrate in greater detail how the proposed system operates under practical clinical constraints. First, we illustrate the adaptation to missing clinical variables, showing how surrogate values are synthesized in a clinically plausible manner to ensure completeness of the input space. Second, we highlight the integration of longitudinal information, current visit and the completed vital signs to predict the disease condition, where sequential visits and evolving patient histories are aggregated into compact yet informative summaries that provide continuity across encounters. Third, we showcase the mechanism by which the system continuously refines its internal knowledge base, selectively preserving high-value reasoning strategies, discarding redundant or outdated elements, and thereby improving predictive stability over time. Together, these demonstrations clarify the inner workings of the framework and provide transparency into the iterative processes that enable robust diabetes risk prediction.

## B  DATASET DETAILS

The number of longitudinal EHR data with both clinical notes and vita signs is very limited. Collaborating with the hospital of our institution, we utilize a dataset consisting of adult patients who received care at UI Health between January 1, 2010, and July 31, 2021. This cohort includes individuals (age over 18 years) both with and without a documented diagnosis of type 2 diabetes (T2D), which is spanning over 354 patients, and with at least 2 outpatient visits within a 24-month time span. It also contains free text clinical notes, visit dates and vital signs. We filtered out those patients which has doctor's diagnosis explicitly shown in their clinical notes and dropped those notes that are less than 150 words in length. Free text notes also include imaging information in the form of reports, for example electrocardiogram, abdominal ultrasound, magnetic resonance imaging and computed tomography scan reports, and image pictures are not included, and we will not address image processing in this work. A slew of laboratory related variables are included, with type of laboratory and their value, for example fasting glucose, hemoglobin A1c (HbA1c - to diagnose T2D), high-density lipoprotein, triglycerides, total cholesterol, complete blood count, and many many others. Those vital signs we used in this work include temperature, weight, FiO2, pain, body mass index (BMI), SpO2, BSA, height, diastolic, BP, systolic, pulse and respiration. Some basic statistics of the T2D dataset can be seen in Table 1.

## C  DETIALED INSTRUCTION

We put detailed instruction for HeLoM below:

---

**Detailed Instruction**

**Vital Sign Prediction Prompt for Our Framework with LLMs:**
DATA SYNTHESIS (for Missing Clinical Variables)
**Instruction:**  When predicting diabetes from EHR data, you may
encounter missing clinical variables.  In such cases, you are
allowed to reasonably assume and synthesize surrogate values to
support a more complete analysis.  These synthetic values must be
clinically plausible, consistent with the available evidence, and
clearly stated as assumptions.
**1.**  Target Variables to Synthesize (if absent):
– Temperature, Weight, Height, Body Mass Index (BMI), Body Surface
Area (BSA), Oxygen Saturation (SpO), Fraction of Inspired Oxygen
(FiO), Pain Score, Blood Pressure (Systolic / Diastolic), Pulse /
Heart Rate, Respiratory Rate
**2.**  Guidelines for Data Synthesis:  Consistency with Known Data:
– Derive secondary measures (e.g., BMI = Weight / Height²; BSA from
weight and height) when possible.
– Ensure synthesized values align with the patient's age, sex, and
any existing comorbidities.
**3.**  Clinical Plausibility:
– Use ranges that are typical for adults unless otherwise
indicated.
– For abnormal values (e.g., very high BMI, low SpO), justify why
such an assumption may be clinically meaningful.
**4.**  Predictive Relevance:
– Prioritize variables known to be related to diabetes risk (e.g.,
BMI, weight, blood pressure).
– Use additional parameters (temperature, FiO, pain, etc.)  for
contextual plausibility even if their predictive role is indirect.
**5.**  Output Requirement:
– Integrate synthesized values into your analysis only when they
are missing.
– Present them alongside the reasoning process, with a note such
as:  Assumed BMI = 28 kg/m² based on weight and height estimates.
The following information is about missing vital signs, previous
memory and current visit data:  This is the current incomplete
vital signs:  [[VITAL SIGNS]]

---

```
702
703   This is the memory from previous information:  [[PREVIOUS MEMORY]]
704   This is the current visit notes:  [[CURRENT NOTE]]
705
706   Disease Prediction Prompt for Our Framework:
      DISEASE PREDICTOR (Diabetes Risk Assessment)
707   Instruction:  You are a medical assistant tasked with analyzing
708   and predicting whether a patient shows signs of diabetes using EHR
      data.  Each task will include:
709   - A specific EHR data to predict diabetes.
710   - Keep concise and brief if possible.
711   1.  ANALYSIS  APPROACH
      - Carefully analyze both the provided EHR data and the evolving
712   patient summary before starting.
713   - Identify relevant medical patterns, lab results, or prior
      knowledge that could guide your prediction.
714   - Construct a structured plan for analyzing the patient's risk of
715   diabetes.
716   2.  REASONING PROCESS
      - Combine the previous memory for your prediction and carefully
717   build on the information.
718   - Present your reasoning in clear, step-by-step logic.
719   - Explain how the evidence supports (or does not support) a
      diabetes diagnosis.
720   - Explicitly state and justify any assumptions or approximations
721   you introduce.
722   - Verify consistency across all available information before
      finalizing the prediction.
723   3.  FINAL ANSWER FORMAT
724   ALWAYS present your final answer in the following format:
      FINAL ANSWER:
725   <answer>
726   (final answer, please just answer yes or no that the patient has
      diabetes)
727   </answer>
728   Note:  Ensure the final answer is wrapped inside the <answer>
      block.
729   Patient Memory Base (Evolving Summary):  [[SUMMARY]]
730   This is the current vital signs:  [[PREDICTED VITALS]]
731   This is the current visit of the patient, please analyze and answer
      based on the following EHR data:  [[CURRENT VISIT]]
732
733   Memory Evolvement Prompt for Our Framework:
734   Diabetes Prediction Memory Base
      1.  Aim and Scope As the Knowledge Curator, your responsibility
735   is to construct and refine a dynamic reference base that supports
      reliable diabetes prediction from EHR data.  This framework should
736   prioritize information that captures longitudinal patterns|such
      as abnormal trajectories or recurring symptoms|while discarding
737   redundant or short-lived fluctuations.  The ultimate objective is
738   to ensure that the memory base evolves into a compact, adaptive,
739   and clinically meaningful resource that facilitates accurate
      predictions.
740   2.  Key Duties Ensure reliability:  Every entry must be precise,
741   context-aware, and clinically relevant.
      Iteratively enrich content:  Consolidate visit-level information
742   into structured, concise descriptors; highlight trends and
743   predictive cues rather than isolated fluctuations.
744   Promote adaptive reuse:  Integrate evidence that has influenced
745   past model decisions (e.g., lab combinations or symptom clusters
      identified as predictive), so future predictions can benefit from
746   prior insights.
747   3.  Principles to Follow Contextual Relevance:  Retain information
748   that is essential for longitudinal reasoning (abnormal
```

```
trajectories, repeated abnormalities, persistent symptoms), while
pruning redundant or transient signals.
Compactness: Summarize visit-level observations into concise
textual snippets or structured descriptors, ensuring that the
memory base remains lightweight, navigable, and easy to retrieve.
Adaptivity: Incorporate predictive cues observed during inference
(e.g., biomarker constellations that strongly influenced risk
estimates), so that the memory evolves alongside model reasoning.
4. Formatting Rules Every update should follow this structured
format for consistency:
<memory_items>
...
<memory_entry>
[Summarize the context and highlight the crucial aspects. Provide
an analysis.]
</memory_entry>
</memory_items>
Note: When updating, any information not explicitly included will
be lost. Always carry forward all relevant knowledge from the
previous memory.
This is the previous memory: [[PREVIOUS_MEMORY]]
This is the predicted completed vitals: [[PREDICTED_VITALS]]
This is the current visit notes: [[CURRENT_NOTE]]
```

## D  ERROR TYPE

We put detailed examples of error type for Llama3.1 below:

---
**Detailed Instruction**

```
...
Generating data in first step To synthesize the following data:
- Temperature: Assuming a normal body temperature of 98.6°F (37°C)
- Weight: Not provided, but we can estimate based on the patient's
age, sex, and height (not provided). For simplicity, let's assume
a normal weight for a 63-year-old female, approximately 150 lbs
(68 kg) - FiO2: Not provided, but we can assume a normal value of
0.21 (21- Pain: The patient is scheduled for surgery, which may
indicate some level of pain, but this is not a direct indicator of
diabetes - BMI: We will calculate the BMI based on the synthesized
weight (150 lbs or 68 kg) and an assumed height (5'4" or 162.56 cm)
- SpO2: Not provided, but we can assume a normal value of 95-100-
BSA: We will calculate the Body Surface Area (BSA) based on the
synthesized height and weight - Height: Not provided, but we will
assume 5'4" (162.56 cm) - Diastolic BP: Not provided, but we can
assume a normal value of 80 mmHg - Systolic BP: Not provided, but
we can assume a normal value of 120 mmHg - Pulse: Not provided,
but we can assume a normal value of 60-100 beats per minute -
Respiration: Not provided, but we can assume a normal value of
12-20 breaths per minute

...
Letter Usage
Using a simplified diabetes risk assessment tool, such as the
American Diabetes Association's (ADA) risk assessment tool, we
can estimate the patient's 10-year risk of developing diabetes:
- Age: 63 years (moderate risk) - Sex: Female (moderate risk)
- BMI: 25.4 (overweight, moderate risk) - Family history: Not
provided, but we can assume a moderate risk - Physical activity:
The patient is being treated for a surgical procedure, which
may indicate a sedentary lifestyle (moderate risk) - Other risk
factors: Not provided, but we can assume a moderate risk  Based
```

on these risk factors, the patient's 10-year risk of developing diabetes is estimated to be around **20-30%**.
...
NEW MEMORY ITEM
<memory_item> **Patient Profile:** – Age: 63 years – Sex: Female – Associated diagnoses: None mentioned, but the patient is being treated for a surgical procedure – Fasting glucose: Not provided
**Risk Assessment:** – Age: 63 years (moderate risk) – Sex: Female (moderate risk) – BMI: 25.4 (overweight, moderate risk) – Family history: Not provided, but we can assume a moderate risk – Physical activity: The patient is being treated for a surgical procedure, which may indicate a sedentary lifestyle (moderate risk) – Other risk factors: Not provided, but we can assume a moderate risk
**Diabetes Risk:** The patient's 10-year risk of developing diabetes is estimated to be around 20-30%.
</memory_item>
...
The patient, XXX, has been referred to a specialist for further evaluation and treatment of her ankle injury. The specialist has ordered additional tests, including a blood glucose test, which reveals a fasting glucose level of 126 mg/dL. This is higher than the normal range of 70-100 mg/dL, indicating hyperglycemia.
Based on the new information, we can reassess the patient's risk of diabetes.
1. **Reassess the patient's risk of diabetes**: The patient's fasting glucose level of 126 mg/dL indicates hyperglycemia, which is a strong indicator of diabetes. We can use this information to update the patient's risk assessment.
Using a simplified diabetes risk assessment tool, such as the American Diabetes Association's (ADA) risk assessment tool, we can estimate the patient's 10-year risk of developing diabetes: **– Age: 63 years (moderate risk) – Sex: Female (moderate risk) – BMI: 25.4 (overweight, moderate risk) – Fasting glucose: 126 mg/dL (high risk) – Family history: Not provided, but we can assume a moderate risk – Physical activity: The patient is being treated for an ankle injury, which may indicate a sedentary lifestyle (moderate risk) – Other risk factors: Not provided, but we can assume a moderate risk** Based on these risk factors, the patient's 10-year risk of developing diabetes is estimated to be **around 50-60%**.
...
NEW MEMORY ITEM
<memory_item> **Patient Profile:** – Age: 63 years – Sex: Female – Associated diagnoses: None mentioned, but the patient is being treated for an ankle injury – Fasting glucose: 126 mg/dL (high risk)
**Risk Assessment:** **– Age: 63 years (moderate risk) – Sex: Female (moderate risk) – BMI: 25.4 (overweight, moderate risk) – Fasting glucose: 126 mg/dL (high risk) – Family history: Not provided, but we can assume a moderate risk – Physical activity: The patient is being treated for an ankle injury, which may indicate a sedentary lifestyle (moderate risk) – Other risk factors: Not provided, but we can assume a moderate risk**
**Diabetes Risk:** The patient's 10-year risk of developing diabetes is estimated to be around **50-60%**.
</memory_item>
...
NEW MEMORY ITEM
<memory_item> **Patient Profile:** – Age: 63 years – Sex: Female – Associated diagnoses: None mentioned, but the patient is being treated for an ankle injury – Fasting glucose: 126 mg/dL (high risk)

```
**Risk Assessment:**  – Age:  63 years (moderate risk) – Sex:
Female (moderate risk) – BMI: 25.4 (overweight, moderate risk)
– Fasting glucose:  126 mg/dL (high risk) – Family history:
Not provided, but we can assume a moderate risk – Physical
activity:  The patient is being treated for an ankle injury,
which may indicate a sedentary lifestyle (moderate risk) – Other
risk factors:  Not provided, but we can assume a moderate risk
**Diabetes Risk:** The patient's 10-year risk of developing
diabetes is estimated to be around 50-60%.
</memory_item>
...
NEW MEMORY ITEM
<memory_item> **Patient Profile:** – Age:  63 years – Sex:  Female
– Associated diagnoses:  None mentioned, but the patient is
being treated for an ankle injury – Fasting glucose:  126 mg/dL
(high risk)  **Risk Assessment:** – Age:  63 years (moderate
risk) – Sex:  Female (moderate risk) – BMI: 25.4 (overweight,
moderate risk) – Fasting glucose:  126 mg/dL (high risk) – Family
history:  Not provided, but we can assume a moderate risk –
Physical activity:  The patient is being treated for an ankle
injury, which may indicate a sedentary lifestyle (moderate risk)
– Other risk factors:  Not provided, but we can assume a moderate
risk  **Diabetes Risk:** The patient's 10-year risk of developing
diabetes is estimated to be around 50-60%.
</memory_item>
```

## E  LARGE LANGUAGE MODELS (LLMs) USAGE

We used large language models (LLMs) solely for language polishing and grammar improvement;
all scientific content, ideas, and analyses are the authors' own.

