# OpenReview forum: "HeLoM: Progressive Disease Detection with Heterogeneous and Longitudinal EHRs via Memory-Augmented LLMs"
_ICLR.cc/2026/Conference — ICLR 2026 Conference Withdrawn Submission_

### Official Review · Reviewer_hKb2 · 2025-11-04

**Soundness:** 2
**Presentation:** 2
**Contribution:** 2
**Rating:** 2
**Confidence:** 4

**Summary:**

This paper proposes HeLoM, a memory-augmented framework for progressive disease prediction that learns from longitudinal electronic health records (EHRs). The method processes patient visit records iteratively, maintaining a memory bank to store and update summaries of past visits, while integrating heterogeneous data (such as clinical notes and vital signs) through LLM-based or interpolation-based imputation strategies. The framework can operate during the inference stage without requiring retraining, and it was evaluated on real-world clinical data comprising 354 patients with diabetes.

**Strengths:**

1. Zero-shot methods adapt to varying patient history lengths, avoiding costly retraining through adaptive memory retrieval and refinement.
2. Addressing data heterogeneity and missing values, systematically comparing LLM interpolation with interpolation methods.
3. Clinically meaningful evaluation metrics, proposing Average Prediction Visit to measure early detection capability.

**Weaknesses:**

1. There is a major issue that needs to be addressed regarding T2D. Most T2DM records are from patients who have already been diagnosed with diabetes when they visit the hosipital. Using these post-diagnosis records directly for prediction is problematic. Instead, records prior to the onset of diabetes should be considered for prediction.

2. In the Related Work section, the definition of 'Memory' is very vague. It should be discussed in relation to existing methods like RAG and what the specific differences are. Additionally, regarding what the authors mentioned: 'LLM. For the current visit, LLMs will utilize the past visits from the "memory bank" that are important for the current prediction as the context.' - what is the actual mechanism?

3. The paper should supplement with experimental results comparing the effectiveness of using RAG methods versus memory, which would better highlight the distinctiveness of using memory in this paper.

4. T2M includes comorbidities, and if memory truncation is applied directly, it may overlook symptoms or behaviors from earlier visits, capturing only the most recent highlights. This is meaningless for diagnosis, as it seems the model is merely guessing whether T2M symptoms have occurred based on the latest key points.

5. The experimental results in Table 2 show anomalies with PromptEHR: Llama3.1 + PromptEHR has a Recall of only 0.057? Also, given that the authors use a large amount of data, they should include fine-tuning methods for comparison.

6. The authors use longitudinal EHR data spanning ten years and employ LLMs for imputation along with linear interpolation. However, formula (2) assumes that vital sign changes are linear, but many physiological indicators exhibit non-linear changes (such as blood glucose). The reasonableness of this assumption is not discussed, and imputation quality assessment and sensitivity analysis are needed.

**Questions:**

If the summarized memory of early visits contains errors, how will these inaccuracies influence the predictions for subsequent visits?

---

> ### Author Response · Authors · 2025-11-21
>
> Thank you for the reviewer’s insightful comment.
> 1. We would like to clarify that our dataset captures the dynamic longitudinal development of each patient prior to the formal diagnosis of T2D. In clinical practice, many patients are diagnosed after a long period of visits. Our EHR sequences naturally reflect this process.
> 2. In our framework, memory refers specifically to a model-generated and model-consumed contextual representation, rather than an external retrieval corpus. For each patient, we leverage the LLM’s own reasoning and summarization capabilities to transform earlier visits into structured memory entries. These memory entries are distilled from the patient’s historical EHR records and formatted into a compact, semantically consistent representation. To validate this distinction, we have already conducted RAG-based experiments.
> 3. See RAG:
> See Rag
> | models | Accuracy | Precision | Recall | F1 Score | AUROC | AUPRC |
> |------|----------|-----------|--------|----------|-------|-------|
> | **Llama3** | **0.573** ★ | 0.543           | **0.921** ★ | **0.683** ★            | 0.500 | 0.502 |
> | **Mediphi** | **0.548**    | **0.840**      | 0.119           | 0.208 | **0.514**           | **0.511** |
> | **Deepseek** | 0.489       | 0.458           | 0.124           | 0.196 | 0.501           | 0.482 |
> | **Mistral** | 0.471            | **0.833** ★ | **0.125**     | **0.217** | **0.547** ★ | **0.640** ★ |
> Also, these is some work which has already show that RAG is not better than the methods of our kind. https://arxiv.org/abs/2510.10454
> 4. To address this issue, we also include PromptEHR as a baseline in our experiments. PromptEHR processes the entire sequence of visits without any truncation or selection, and feeds all historical records directly into the LLM. This provides a comprehensive “full-history” comparison that ensures no clinically relevant information is lost.
> 5. After the check, we found at that point the experiments did not finish, I am trying to finish the experiment soon. Based on the result we have so far, the result has high chance that is below our methods.
> 6. Core biomarkers used in longitudinal diabetes management, such as hemoglobin A1c (HbA1c), naturally possess strong smoothing properties. Physiologically, HbA1c functions as a low-pass filter — it inherently filters out acute, short-term glycemic fluctuations (i.e., the ‘spikes’ that the reviewers are concerned about).https://www.ccjm.org/content/83/5_suppl_1/S4. https://pmc.ncbi.nlm.nih.gov/articles/PMC4933534/. https://pmc.ncbi.nlm.nih.gov/articles/PMC6074595/.

---

> > ### Author Response · Authors · 2025-11-23
> >
> > Dear reviewer, the complement experiment for PromptEHR is:
> > without vitals:
> >     Accuracy: 0.5763
> >     Precision: 0.7077
> >     Recall: 0.2599
> >     F1 Score: 0.3802
> > with vital:
> >     Accuracy: 0.5743
> >     Precision: 0.7188
> >     Recall: 0.2599
> >     F1 Score: 0.3817

---

> ### Comment · Reviewer_hKb2 · 2025-11-26
> **Response to author’s rebuttal**
>
> I appreciate the author's response to my concerns; the framework is indeed easy to understand. However, I maintain my original rating due to the lack of rigorous experimental evidence demonstrating the actual benefits of the memory mechanism. For instance, while the paper employs dynamic memory for T2D prediction and memory updating, it fails to test different memory lengths or updating strategies. As Reviewer mCnm pointed out, it remains confined to the prompt engineering method. More critically, it should incorporate blood glucose level predictions across different age groups, modeling the higher risks associated with the elderly, and compare its performance against models for younger populations.

---

### Official Review · Reviewer_mCnm · 2025-11-08

**Soundness:** 2
**Presentation:** 2
**Contribution:** 2
**Rating:** 2
**Confidence:** 5

**Summary:**

To tackle three persistent challenges, inefficient utilization of previous visits, overlook of EHR's inherent nature and the missingness of vital signs, this paper proposes HeLoM to adaptively fetch previously refined memory, incorporate vital signs and utilize two imputation strategies to handle missing data.

**Strengths:**

- The motivation is clear.
- The paper is easy to read and understand.

**Weaknesses:**

- Limited novelty. The core contributions—using a memory mechanism introduced in prior work and incorporating vital signs with standard imputation during prediction—are incremental. As presented, the methodological novelty does not meet the typical bar for ICLR.
- Insufficient experimental analysis. The paper lacks comprehensive experiments to substantiate the claims. More thorough evaluations are needed to demonstrate the method’s effectiveness and robustness.

**Questions:**

- Novelty is limited. The manuscript adopts a memory-augmented inference paradigm already explored in prior work, and the two imputation strategies are standard. As implemented, the pipeline reads more like pre-/post-processing around a base model than a new framework. Please clarify the distinct technical contribution.
- No validation of the generated memory. Provide a quantitative validation of memory quality: e.g., factual consistency checks, information coverage/omission rates. Report correlations.
- “Complementary” imputations. "Two complementary imputation strategies", explain in what sense the two imputation methods complement each other.
- Scope of evaluation. A single dataset/task is insufficient to demonstrate generality. Please include additional EHR datasets (e.g., MIMIC-III/IV) and multiple tasks.
- Positioning vs. few-shot LLMs and retraining baselines. Clarify the comparison target: if the claim is “test-time adaptation via contextual enrichment,” focus baselines on prompt-based and retrieval-augmented methods. "Unlike approaches that rely on continuous retraining, our method performs test-time adaptation purely through contextual enrichment." Here the "approaches" are deep learning methods? Your method targets at LLMs, where prompt engineering does not need model retraining. Why compare with deep learning methods?
- Baseline coverage. Current comparisons (two baselines) are not enough. Adding general-domain and medical-domain baselines is needed.
- Backbone selection & metrics. The backbones should vary with its targeted domain and model size. In the medical setting, report AUROC and AUPRC (AUPRC preferred for class imbalance). Since LLMs can output probabilities, evaluate on calibrated scores rather than only text outputs.
- Statistical testing. Including confidence intervals and statistical significance will validate the effectiveness of the method.
- The experiment "AVERAGE VISIT INDEX FOR DISEASE DETECTION" does not compare with the baselines. So what is the point of this experiment? Limited by context length? There are LLMs with enough context length.
- Model-specific claims. Statements such as "The exception is for Llama3.1, which is partly due to its limited capabilities in handling heterogeneous data." must be supported with objective evidence.

---

> ### Author Response · Authors · 2025-11-21
>
> 1. We introduce a task-specific dynamic memory construction mechanism rather than a static retrieval module.Most existing memory-augmented LLM methods rely on static prompting or retrieval (e.g., RAG-style). In contrast, our framework constructs a self- evolving memory that is explicitly aligned with the temporal and multimodal characteristics of EHR data.
> Our two imputation strategies are integrated as part of the self-evolving memory reasoning pipeline. The memory is conditioned on the model’s evolving predictions, and missingness is addressed adaptively rather than with a fixed prior. Our contribution is a unified inference framework, not a set of isolated pre/post-processing techniques.
> 2. Sorry for the confusion, actually our two imputation strategies are two independent methods to do the imputation. They are not sequential relationships.
> 3. For non-training methods, we already include several prompt-based baselines using different prompting strategies. Our experiments compare against multiple prompting paradigms, a fair evaluation for the training-free LLM methods.
> Because our method is fully training-free, our baselines are also chosen to be training-free.A major intended use case of our framework is scenarios where fine-tuning is infeasible (e.g., limited labeled EHR data, rapid deployment, or restricted compute). Therefore, all primary baselines with including prompting and RAG variants that we just conducted,use existing models purely at inference time without any retraining. This aligns with the goal of evaluating test-time adaptation in a fair and controlled setting.
> Few-shot fine-tuning is not feasible due to the difficulty of EHR data collection. Our dataset is clinically meaningful but very costly to collect. At the current stage, the dataset size is not sufficient to support few-shot of LLMs.
> 4. In addition to two baselines, we add an PromptEHR. And here we just have the RAG-based method’s result. See:
> | models | Accuracy | Precision | Recall | F1 Score | AUROC | AUPRC |
> |------|----------|-----------|--------|----------|-------|-------|
> | **Llama3** | **0.573** ★ | 0.543           | **0.921** ★ | **0.683** ★            | 0.500 | 0.502 |
> | **Mediphi** | **0.548**    | **0.840**      | 0.119           | 0.208 | **0.514**           | **0.511** |
> | **Deepseek** | 0.489       | 0.458           | 0.124           | 0.196 | 0.501           | 0.482 |
> | **Mistral** | 0.471            | **0.833** ★ | **0.125**     | **0.217** | **0.547** ★ | **0.640** ★ |
> Also, these is some work which has already show that RAG is not better than the methods of our kind. https://arxiv.org/abs/2510.10454
> 5. Our dataset is strictly protected under institutional compliance policies and can only be accessed and processed within a secure, offline clinical computation environment. This environment permits running only a single 40GB A100 GPU. Due to these restrictions, larger LLMs are currently not feasible.
> | Models | Methods | AUROC | AUPRC |
> |------|------|-------|-------|
> | **DeepSeek** | BL-16K | 0.579 | 0.554 |
> | | BL-24K | 0.590 | 0.564 |
> | | PromptEHR | **0.672**  | 0.614 |
> | | HeLoM w. LLM's V | **0.707** ★ | **0.649** ★ |
> | | HeLoM w. Interp's V | 0.670 | **0.616**  |
> | | PromptEHR w/o V | **0.672**  | 0.614 |
> | | HeLoM w/o V | 0.661 | 0.613 |
> | **Mistral** | BL-16K | 0.672 | 0.640 |
> | | BL-24K | 0.658 | 0.633 |
> | | PromptEHR | 0.619 | 0.587 |
> | | HeLoM w. LLM's V | **0.722** ★ | **0.673** ★ |
> | | HeLoM w. Interp's V | **0.711**  | **0.666**  |
> | | PromptEHR w/o V | 0.619 | 0.587 |
> | | HeLoM w/o V | 0.682 | 0.656 |
> | **Llama3.1** | BL-16K | 0.590 | 0.561 |
> | | BL-24K | 0.607 | 0.579 |
> | | PromptEHR | 0.497 | 0.498 |
> | | HeLoM w. LLM's V | **0.716**  | **0.655** |
> | | HeLoM w. Interp's V | 0.700 | 0.640 |
> | | PromptEHR w/o V | 0.497 | 0.498 |
> | | HeLoM w/o V | **0.726** ★ | **0.667** ★ |
> | **MediPhi** | BL-16K | 0.610  | **0.568**  |
> | | BL-24K | 0.590 | 0.554 |
> | | PromptEHR | 0.524 | 0.506 |
> | | HeLoM w. LLM's V | **0.615** ★ | 0.565 |
> | | HeLoM w. Interp's V | **0.610** | **0.571** ★ |
> | | PromptEHR w/o V | 0.515 | 0.502 |
> | | HeLoM w/o V | 0.589 | 0.561 |
> 6. This is strictly not an  experiment, but as an analysis of temporal dependency. Its goal is to quantify how long and early in the patient trajectory difference become detectable.
> 7. In fact, in the 4.5 part of our study, we do provide a detailed examination of Llama 3’s behavior on this task. This includes both quantitative evaluation and qualitative inspection of its generated outputs under heterogeneous EHR inputs.

---

> > ### Comment · Reviewer_mCnm · 2025-11-26
> >
> > I appreciate the authors’ efforts in addressing my comments. I will retain my original score. I encourage the authors to further strengthen the methodology and experiments in future revisions.

---

### Official Review · Reviewer_uGpK · 2025-11-09

**Soundness:** 3
**Presentation:** 3
**Contribution:** 3
**Rating:** 6
**Confidence:** 2

**Summary:**

This paper aims to address (early) disease detection in longitudinal electronic health records (EHRs) using a memory-augmented large language model (LLM) framework. The proposed approach, called HeLoM, iteratively processes a patient's visits and dynamically incorporates relevant prior visits as memory context for disease prediction (demonstrated on Type-2 diabetes). It integrates heterogeneous data by including structured vital sign measurements alongside clinical notes. To handle missing values in vital signs, two complementary imputation strategies are used: an LLM-based imputation and linear interpolation. Experiments on a newly collected 10-year EHR dataset show improved early detection and higher recall/F1 scores over baseline methods.

**Strengths:**

- From my personal perspective, the proposed work seems novel and highly relevant to both healthcare and llm communities. This work proposes a inference framework that allows an LLM to handle long-term patient histories adaptively. Unlike prior methods that truncate or fix the number of visits, the iterative memory bank enables the model to dynamically incorporate relevant context from arbitrary-length EHR sequences without fine-tuning. Integrating structured vital sign data with unstructured notes via prompt engineering further sets this approach apart from existing studies that focus solely on clinical text.

- Well-motivated. The heterogeneity-aware prompting and dual imputation strategies are well-motivated, and strengthen the work robustness to irregular data.

- The experimental evaluation is comprehensive. And results look good

**Weaknesses:**

- Perhaps a straightforward weakness is that the study is evaluated on a single institution’s dataset for one disease (Type-2 diabetes) (I agree this is important and meaningful data), which may limit the generality of the conclusions.

- The experimental comparison focuses on prompting-based LLM approaches, maybe including conventional or fine-tuned models for EHR disease prediction would be more thorough and interesting

- Statement of ethics seems missing

**Questions:**

Please see weaknesses above

**Details Of Ethics Concerns:**

The work included a new collected patients dataset from authors' institutions, but no ethics statement is included. Hopefully we could see they add them during discussion phase

---

> ### Author Response · Authors · 2025-11-21
>
> Thank you for the reviewer’s suggestion.
> 1. The choice of a single-institution longitudinal dataset is driven by real clinical constraints. The methodological contribution of HeLoM is training-free inference, not dataset-specific tuning.
> 2.
> 1）We have already included strong domain-specific baselines. Our experiments incorporate Microsoft’s open-source medical model, which is a widely used and highly competitive foundation model for clinical tasks.
> 2) ​​Fine-tuning large medical models is not directly comparable to our intended scope.
> Our goal is to evaluate whether generalizable untrained prompting strategies can leverage latent medical knowledge without requiring task-specific training.
> 3) Fine-tuning also requires substantially larger datasets than ours. Our EHR cohort is not yet large enough to support stable fine-tuning of medical LLMs.
>
> Details Of Ethics Concerns:
> Privacy is a major concern in a variety of AI applications that operate on identifiable data; healthcare data has been protected by the HIPAA law. The data for this project will create a HIPAA limited dataset where the direct identifiers will be removed.

---

### Official Review · Reviewer_rQLu · 2025-11-16

**Soundness:** 3
**Presentation:** 2
**Contribution:** 2
**Rating:** 2
**Confidence:** 5

**Summary:**

Thank you for your great ideas on incorporating the memory-augmented inference-based framework, handling missing data, and especially, integrating multimodal heterogeneous data sources to enhance disease prediction (T2D).

**Strengths:**

Comprehensive analysis from the author with a different LLM backbones

**Weaknesses:**

Unfortunately, I find that the current stage of the work needs to be improved in the two major points:
First, the key point of this work is the incorporation of memory-augmented inference-based methods; however, it lacks a comprehensive analysis of the effectiveness of HeLoM in terms of the time elapsed from time t to the previous memory point, as well as the effectiveness of memory as a function of temporal distance. Specifically, the proposed HeLoM framework does not evaluate how performance varies with different memory spans, e.g., from very recent contexts (t–1 to t–5) to moderately distant (t–5 to t–10) or long-term memory (t–10 and beyond). This leads to two critical gaps:
i. Temporal sensitivity is not quantified: No ablation or sensitivity analysis examines how the elapsed time between the current timestep t and the retrieved memory point influences model performance.
ii. The memory–performance trade-off is not explored: Without experimental evidence showing performance gains as memory grows, it is unclear whether long memory windows are beneficial or if diminishing returns occur after a particular horizon.

Second, even a comprehensive analysis from the author with a different LLM backbone; however, it was run only on a single dataset. The reviewer wonders what the findings will be for other datasets.


Minor:
1. Please provide a short description of the ethical approval for the obtained dataset (even if it is with anonymous information)
2. A trade-off between evaluated metric performance and computational analysis should be made, so that the reader can expect to consider the limitations of the shared GPU from the hospital, including the time it takes to conduct the experiments.
3. Detailed hyperparameter setting up should also include to support the reproducibility analysis.

**Questions:**

Besides that, the following concerns need to be discussed or provided with more clarification or justifications:
1. Please provide a table that shows the comprehensive advantages and disadvantages of the proposed approach (HeLoM) compared to the discussed SOTA works.
2. Page 7: Is there any analysis to justify why HeLoM improves Recal compared to PromptEHR?
The authors confirm: “These findings suggest that the principled design of HeLoM —memory augmented, heterogeneity, and missingness handling – enables it to achieve a more clinically desirable balance between sensitivity and reliability, offering more consistent identification of high-risk patients compared to baseline methods.”
Unfortunately, it is not clear enough to conclude with a convincing justification. Could you please provide further clarification on this point?
3. Section 4.3. Missing values: what happens if the missing values for the vital signal at time t are significantly different from the previous times in case of an emergency visit ===> This will lead to a strong bias in the prior vital sign, especially in terms of the Interpolation approach, which strongly relies on recent and prior visit results.
4. It is quite confusing from the obtained results from Fig. 2 vs Table 4? Fig. 2 confirms that linear interpolation outperforms the LLM-based imputation. Please provide clarification for these points; otherwise, it leads to a contradiction between the findings from the two experiments. And if it is the case for linear interpolation, I will return to the question raised in point 3 above.
If necessary, please also provide the complete results from linear imputation, as shown in Table 4 for LLM-based imputation.

---

> ### Author Response · Authors · 2025-11-21
>
> Thank you for the reviewer’s suggestion.
> 1. Our dataset is strictly protected under our institutional policies and can only be accessed and processed within, offline clinical computation environment. This environment only has a single 40GB A100 GPU. In this case, our total spent hours for the testing dataset is about 6-8 hours.
> 2. Our parameters are: temp: 0; top p: 1; top k: -1. The rest are default, good for reproducibility;
> 3. Unlike DL-based methods requiring supervised retraining, HeLoM performs pure test-time contextual adaptation without updating model parameters. HeLoM constructs a patient-specific temporal memory through LLM-generated summaries. Ours combines semantic LLM imputation with structured continuous-value interpolation, while existing LLM baselines (e.g., PromptEHR) input full sequences.
> 4. Memory-based temporal abstraction reduces noise and amplifies early risk patterns. Instead of passing the raw full history, HeLoM generates memory summaries that highlight clinically meaningful deviations. Explicit handling of heterogeneity and missingness avoids false negative
> 5. Core biomarkers used in longitudinal diabetes management, such as hemoglobin A1c (HbA1c), naturally possess strong smoothing properties. Physiologically, HbA1c functions as a low-pass filter — it inherently filters out acute, short-term glycemic fluctuations (i.e., the ‘spikes’ that the reviewers are concerned about).https://www.ccjm.org/content/83/5_suppl_1/S4. https://pmc.ncbi.nlm.nih.gov/articles/PMC4933534/. https://pmc.ncbi.nlm.nih.gov/articles/PMC6074595/.
> 7. Sorry for the confusion. In 4.4, table 4 is used to explain in the Impact of Data Imputation and  Impact of Incorporating Vital Signs. And the Fig. 2 is used to explain the part of Impact of Data Imbalance Ratio. For the table 4, the baseline BL-24K and PromptEHR with LLM-generated vital signs should be compared with BL and PromptEHR in the table, and the HeloM with original incomplete vital signs should be compared with the HeLoM with generated vital signs and without vital signs. For the fig.2, this is an evaluation under the imbalance situation. We down sample the positive cases with the LLM-generated method as an example to explain the effectiveness for real-world cases.

---

### Public Comment · ~Sihang_Zeng1 · 2025-11-12
**Discuss a concurrent work**

Hi authors,

Thank you for the interesting work done in this paper. We have a concurrent work that also uses long-term memory to reason over longitudinal EHR data. The paper is accepted by GenAI4Health workshop at NeurIPS 2025 (https://arxiv.org/abs/2510.10454). The imputation module in your paper looks interesting and novel, and it’s good to see this idea works on T2D data.

IMHO, one difference between your work and ours is that we preserve the heterogeneity in EHR for generalizability and missingness is considered as a normal phenomenon which lefts to LLM to determine whether it’s informative or not, while your work demonstrates the importance of value imputation. It would be appreciated if you could discuss the relevance and differences with our paper, especially on how the imputation adds value to the framework. I believe this could improve the understanding of the heterogeneity in EHR and advance the field of longitudinal EHR modeling.

Thanks again for your work and look forward to hearing from you!

---

### Note · Authors · 2026-01-09

**Comment:**

The authors wish to withdraw this submission.

**Withdrawal Confirmation:**

I have read and agree with the venue's withdrawal policy on behalf of myself and my co-authors.